# Flow as the Cross-Domain Manipulation Interface

**Mengda Xu** [1, 2, 3]    **Zhenjia Xu** [1,2]    **Yinghao Xu** [1]    **Cheng Chi** [1,2]
**Gordon Wetzstein** [1]    **Manuela Veloso** [3,4]    **Shuran Song** [1,2]
[1] Stanford University, [2] Columbia University,
[3] J.P. Morgan AI Research, [4] Carnegie Mellon University
https://im-flow-act.github.io

**Abstract:** We present Im2Flow2Act, a scalable learning framework that enables robots to acquire real-world manipulation skills without the need of real-world robot training data. The key idea behind Im2Flow2Act is to use object flow as the manipulation interface, bridging domain gaps between different embodiments (i.e., human and robot) and training environments (i.e., real-world and simulated). Im2Flow2Act comprises two components: a flow generation network and a flow-conditioned policy. The flow generation network, trained on human demonstration videos, generates object flow from the initial scene image, conditioned on the task description. The flow-conditioned policy, trained on simulated robot play data, maps the generated object flow to robot actions to realize the desired object movements. By using flow as input, this policy can be directly deployed in the real world with a minimal sim-to-real gap. By leveraging real-world human videos and simulated robot play data, we bypass the challenges of teleoperating physical robots in the real world, resulting in a scalable system for diverse tasks. We demonstrate Im2Flow2Act's capabilities in a variety of real-world tasks, including the manipulation of rigid, articulated, and deformable objects.

**Keywords:** Robots, Learning, cross-domain, cross-embodiment

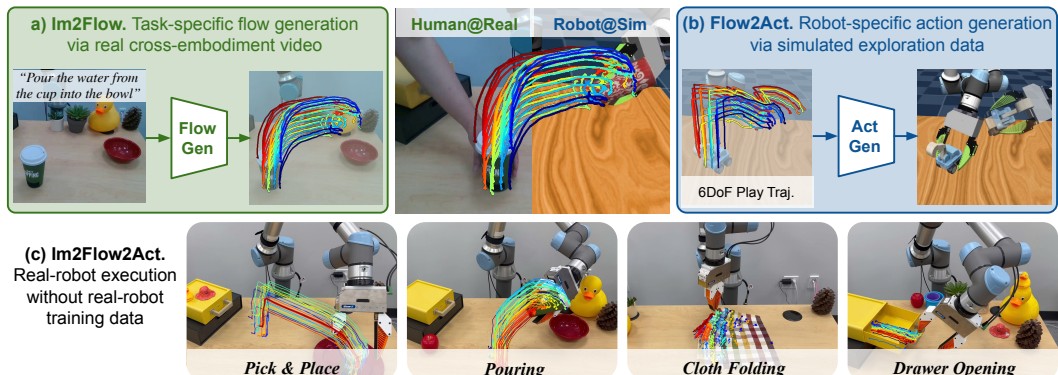

Figure 1: **Flow as the Cross-domain Manipulation Interface.** In Im2Flow2Act, we utilize object flow to bridge the domain gap between both embodiments (human v.s. robot) and training environments (real v.s. simulation). Our final system is able to leverage both **a)** action-less human video for task-conditioned flow generation and **b)** task-less simulated robot data for flow conditioned action generation, results in **c)** a language-conditioned multi-task system for real-world manipulation tasks but without the need of real-robot training data.

## 1   Introduction

A key step for scaling up robot learning is giving robots the ability to learn from various data sources beyond expensive real-world robot data [1, 2, 3, 4, 5]. Prior work has approached this problem from

8th Conference on Robot Learning (CoRL 2024), Munich, Germany.

two main directions: learning from cross-embodiment videos [6, 7, 8, 9, 10, 11, 12, 13] and simulated robot data. While both approaches have shown promising progress, they each face significant challenges. Learning from cross-embodiment data is hampered by the large embodiment gap, while simulated data struggles with a significant sim-to-real gap and the complexity to build task-specific environment, especially for contact-rich manipulation tasks.

In this paper, we introduce **Im2Flow2Act**, a novel framework to learn real-world manipulation skills but without the need of real-world robot training data. Our key idea is to use object flow—the exclusive motion of the manipulated object, excluding any background or embodiment —as a unifying interface to connect human videos and simulated data, and achieve one-shot generalization for new skills in the real world. Our framework (Fig. 1) consists of two main components: a flow generation network (Fig. 1-a) trained on **cross-embodiment demonstration videos** and a flow-conditioned policy (Fig. 1-b) trained on embodiment-specific **cross-environment simulated data**. During inference, the flow generation model generates the complete task-specific object flow based on the initial visual observation and task description (Fig. 1-c). Generating a complete object flow helps to minimize the visual appearance gap between the training and inference when learning from cross-embodiment demonstration. Conditioned on this generated flow, our task-agnostic policy generates actions to execute the task. We outline the benefits of using object flow as a unifying interface:

- **Expressive task descriptions:** Object flow not only captures changes in object pose but also accounts for articulations and deformations. Its versatility enables it to represent a wide range of objects and tasks, including rigid, articulated, and deformable objects. In Im2Flow2Act, we leverage this versatility by training a single manipulation policy for diverse tasks.
- **Embodiment agnostic:** Object flow describes the change in the state of an object caused by an action rather than the action itself, making it independent of the agent's embodiment. Compared to prior work [14] utilizing uniform grid flows that also track the embodiment motion, our method uses object flow, making our framework more effective for cross-embodiment learning.
- **Minimal sim-to-real gap:** Compared to image-based representations, flow focuses on motion rather than appearance, reducing the sim2real gap and aiding generalization. Unlike prior works utilizing flow for manipulation that rely on heuristic policy [15], realworld teleoperation data [14, 16], or both [17], our closed-loop policy is entirely learned from simulated robot exploration data generated by a set of predefined action primitives.
- **Interpretable:** Unlike vector-based state representations, flow-based representation has intuitive physical meaning. This makes it a good interface for human-robot interaction. For example, a user can easily understand and select a flow when multiple flows exist for a task.

To utilize object flow as an interface to learn from diverse data, our system contains two components:

- **Flow generation network:** The goal of the flow generation network (Fig. 1-a) is to learn high-level task planning through cross-embodiment videos, including those of different types of robots and human demonstrations. We develop a language-conditioned flow generation network built on top of the video generation model Animatediff [18]. Compared to prior approaches [14, 15, 17] that train on the original flow space, we leverage the autoencoder (AE) from Stable Diffusion [19] to first compress the flow into latent space, and then train the model on that compressed representation, making our training process more efficient.
- **Flow-conditioned policy:** The goal of the flow-conditioned imitation learning policy (Fig. 1-b) is to achieve the flows generated by the flow generation network, focusing on low-level execution. The policy learns entirely from simulated data to build the mapping between actions and flows. Unlike most sim-to-real work that requires building task-specific simulation, our policy learns entirely from exploration data generated by a set of predefined primitive actions, which is easier to collect and more diverse. Since flow represents motion, a common concept across both real-world and simulation, our policy can be seamlessly deployed in realworld settings.

Together, the flow generation network and the flow-conditioned policy form a cohesive system. Our final system provides a scalable framework for acquiring robot manipulation skills by bridging cross-embodiment demonstration video and cost-effective simulated robot data via object flows as

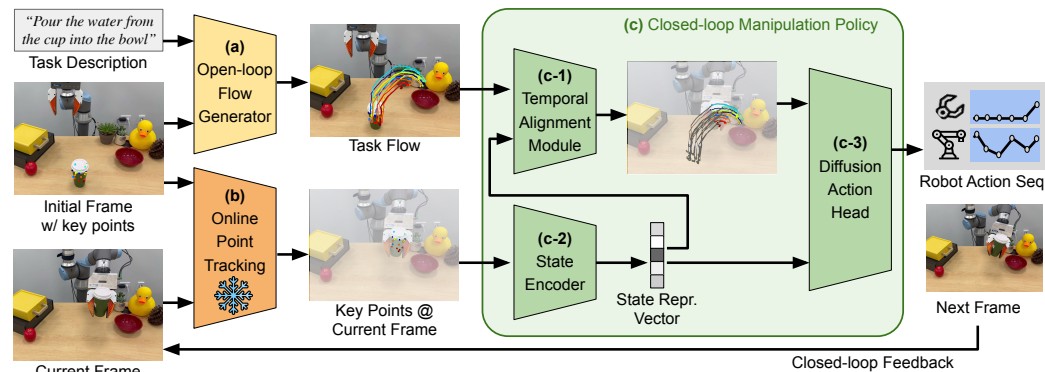

Figure 2: **Im2Flow2Act Overview.** Given a task description and the initial frame, the flow generator (**a**) generates a complete object flow for the task (i.e., task flow). The closed-loop manipulation policy (**c**), takes in the task flow and the keypoint location at the current frame to infer the robot actions. Within the manipulation policy, a temporal alignment module (**c-1**) is used compare task flow and keypoint location at the current frame to infer remaining task flow (**flow in color**) for the diffusion head (**c-3**) to generate actions.

an interface. We achieve an average success rate of 81% across four real-world tasks, including those involving rigid, articulated, and deformable objects without any real-world robot data for training.

## 2 Related Work

**Flow-based manipulation.** Robot policies leveraging flows have demonstrated promising results in manipulating articulated objects [20, 21] and tools [22]. However, these approaches are limited to specific tasks. With recent advances in point tracking algorithms in computer vision, flows can be estimated in a video sequence through learning-based methods. Vecerik et al. [16] achieved few-shot learning through tracking flows but required explicit pose estimation and was still limited to rigid objects. To make flow applicable to more general manipulation tasks, ATM [14] learned a flow-conditioned behavior cloning policy that can be applied to both articulated and deformable objects. However, ATM still required collecting in-domain robot data through teleportation in the real world. Yuan et al. [15] proposed a heuristic flow-based policy but required manual positioning of the robot gripper on the object, making the overall system less autonomous. More recently, Bharadhwaj et al. [17] propose to learn a residual policy on top of the heuristic-based policy but still require collecting real-world robot data. In Im2Flow2Act, we train a data-efficient and fully autonomous flow-conditioned policy from task-less simulation datasets which does not require costly in-domain robot data collection and can be transferred to realworld in one-shot.

**Learning from cross-embodiment data.** A large body of work [23, 24, 25, 26] has studied leveraging cross-embodiment data to learn robotic policies. Prior works explore different directions, including visual pretraining [8, 27, 28], learning reward functions [7, 29, 30, 31], extracting affordances [32, 9, 33, 34], hand pose detection [6, 35, 36], and domain translation [37, 38, 39, 40, 41, 42, 43]. As most of cross-embodiment data lacks explicit action or the action is challenging to transfer due to embodiment gaps, many prior works still require collecting in-domain robot data [44, 42, 12, 11, 45] to mitigate the large embodiment gap. A number of works [46, 14, 15] have learned video generation or flow generation models through cross-embodiment data. ATM [14] demonstrates promising results in learning flow generation from human demonstrations. However, as ATM generates grid flow that inevitably captures embodiment motion, it requires costly real-world robot data to train alongside the human demonstration data. This ensures the generated flow aligns with the distribution needed for robot manipulation policies. Im2Flow2Act predicts only the flow of objects, independent of the embodiment. This approach allows us to seamlessly learn from human videos without needing any in-domain robot data. Please see Appendix A for more related works.

## 3 Method

We aim to build a framework that leverages object flow as a unified interface for robots to acquire real-world manipulation skills but without the need of costly real-world in-domain robot data. Our key idea is to use object flows as a general and expressive interface to bridge:

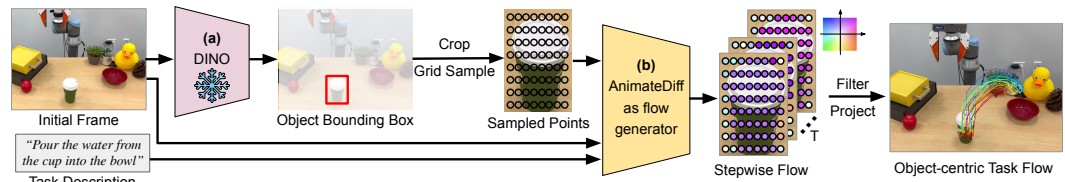

Figure 3: **Flow Generation Network.** The flow generation network outputs *object* flow for the *complete* task conditioned on the initial frame and task description. The object of interest is first detected using Grounding DINO (**a**), then we sample grid points inside the bounding box as the initial keypoints. AnimateDiff (**b**) takes in keypoints and task description to generates the future flow. We postprocess the generated flow through a motion filter to obtain an object-centric task flow.

**Cross-Embodiment data:** Transferring actions between different embodiments is often hindered by the embodiment gap. However, object flow encapsulates transferable task knowledge independent of embodiment-specific actions. To exploit this, we develop a flow generation network focused solely on extracting object flows from cross-embodiment videos.

**Cross-Environment data:** In this work, cross-environment refers to simulation and real-world environments. Visual inconsistencies, such as differences in scene backgrounds and object textures, pose challenges in transferring learned policies from simulation to real world. Conversely, object flow captures motion dynamics—changes in object pose, articulation, and deformation—consistent across both simulated and real world, providing an ideal interface for policy learning. Our flow-conditioned imitation learning policy is learned entirely from the diverse simulated exploration data generated by a set of predefined primitive actions, which fosters an understanding of the relationship between actions and resulting object flows, further reducing the need for task-specific environments.

During inference, the flow generation network first generates a complete task flow at the beginning of the task, and then the policy executes the actions to reproduce the generated flow.

## 3.1 Flow Generation Network

Our main idea is to form permutation-invariant object flows into a sequence of structured rectangular flow images. This approach allows us to better utilize pretrained image and video generation models for more efficient training. Specifically, we use Grounding DINO [47] to obtain the bounding box of the object of interest from the initial RGB frame, followed by uniform sampling within the box, resulting in a rectangular flow $\mathcal{F}_0 \in \mathrm{R}^{3 \times H \times W}$. The first two channels represent the $u, v$ coordinates of object keypoints in image space, while the third represents their visibility. For tasks involving multiple objects, we first obtain the bounding boxes of all relevant objects and perform uniform sampling within each. The number of keypoints sampled is proportional to the area of each bounding box. We then apply TAPIR [48] to the videos, generating the rectangular flow sequence $\mathcal{F}_1 \in \mathrm{R}^{3 \times T \times H \times W}$ with $T$ steps in the temporal space. With this flow representation, we leverage the diffusion-based video generation architecture AnimateDiff [18] as our flow generator network, conditioning it on the initial RGB frame and task description to generate object flow. The generated flow is processed by motion filters to retain only keypoints on the object, producing a complete object-centric task flow. The flow generation pipeline is illustrated in Fig. 3.

**Complete Object-Centric Task Flow**. Im2Flow2Act generates a complete object-centric task flow, contrasting with the immediate next-step grid flows in ATM [14]. This design minimizes the embodiment gap between human demonstration and robot execution. First, object-centric flows reduce motion distractions from the agent. Grid flows predominantly capture the embodiment's motion (e.g., human arm), leading to out-of-distribution flow generation for the robot policy, which is trained with robot-specific flows. As a result, ATM requires additional real-world robot data to close this gap, whereas our method directly learns flow generation from human demonstrations. Second, generating a complete flow from the initial frame avoids visual gaps during execution. Immediate next-step flows are negatively impacted by the embodiment gap, as humans often contact and occlude objects during demonstrations. This conditions the model on visual input where the embodiment and objects are closely attached. However, during robot deployment, the gripper will contact objects, leading to

out-of-distribution visual input. Generating a complete flow from the initial frame avoids this, as the embodiment need not be in contact with objects.

**Compressing Flow into Latent Space**. High-precision flow is critical for enabling robots to achieve fine-grained control over objects. However, generating object flow at high resolution introduces significant computational complexity and results in inefficient generation speed. Inspired by previous work [19], we propose to compress the object flow into a compact latent space with a lower spatial dimension, followed by training the generative model within this latent space. StableDiffusion (SD) [19] compresses input images using an autoencoder (AE) proposed in VQGAN [49], resulting in a well-distributed latent space by training on a vast amount of RGB images. In this work, we explore the idea of using AE to encode the flow. To leverage this well-structured latent space, we fix the encoder $E_\phi(\cdot)$ from the AE and finetune the pretrained decoder $D_\theta(\cdot)$ to better adapt it to the flow images. The latents $x_{1:T} \in \mathrm{R}^{C \times T \times H_x \times W_x}$ for input flow can be obtained by $x_{1:T} = E_\phi(\mathcal{F}_i)|i \in [1, T]$, where $H_x$ and $W_x$ denote the spatial dimensions, which are downsampled by a factor of 8 compared to the input flow spatial dimensions $H$ and $W$.

**Video Diffusion Models for Flow Generation**. As we share the same latent space as StableDiffusion (SD), we can utilize the pretrained SD to provide a strong content prior for flow generation. Therefore, we choose to inflate the SD network along the temporal axis for flow generation rather than training a model from scratch. We then insert the motion module layer into SD proposed by Animatediff [18] to model the temporal dynamics for flow generation. The motion module performs self-attention on each spatial feature along the temporal dimension to incorporate temporal information The SD model with the motion module, learns to capture the temporal variations in keypoints' location $(u, v)$ in the image space. We train the motion module layer from scratch but only insert LoRA (Low-Rank Adaptation) layers [50] into the SD model.

## 3.2 Flow-Conditioned Imitation Learning Policy

Our imitation learning policy $p(\mathbf{a_t}|\mathcal{F}_{0:T}, s_t, \rho_t)$ takes the object flow $\mathcal{F}_{0:T}$ for a complete task (i.e., task flow), the current state representation $s_t$, and the robot proprioception $\rho_t$ as input. The output is a sequence of actions $\{a_t, \ldots, a_{t+L}\}$ of length $L$, starting from the current time $t$, denoted as $\mathbf{a_t}$. The robot action $a_t$ includes a 6-DOF end-effector position in Cartesian space and a 1-DOF gripper open/close state. The state representation $s_t$ is formed by keypoints location at current frame denoted as $f_t$ containing $N$ keypoints' locations $\{(u_n^t, v_n^t)\}_{n=1}^N$ in the image space at time $t$ and the 3D coordinates of $N$ keypoints $x_0 \in R^{N \times 3}$ at the initial frame.

During inference, the task flow $\mathcal{F}$ is generated once by the flow generation network at the beginning of the task, as described in Sec. 3.1. We leverage an online point tracking algorithm to obtain the $f_t$ during robot manipulation for the same set of keypoints. The policy consists following components: a state encoder $\phi$ takes the $N$ keypoints encapsulate in the $f_t$ and their corresponding initial 3D coordinates $x_0$ as input to generate state representation $s_t$, a temporal alignment module $\psi$ which compares task flow and current keypoints location $f_t$ to infer remaining task flow, and finally, a diffusion action head [51] to output the robot action.

**Training data:** We briefly describe how we construct the training data and some decision choice (See supplementary for more details). For an episode with time duration of $T'$, we run the point tracking algorithm to get location of keypoints on object being manipulated at each step, i.e, $f_{0:T'}$. We can get a dataset $\mathcal{D}$ in the form of trajectories $\tau_k = \{(\rho_0, f_0, a_0), ..., (\rho_{T'}, f_{T'}, a_{T'})\}$. During the training, we randomly sample $T$ frames among $f_{0:T'}$ to treated it as task flow $\mathcal{F}_{0:T}$.

**State Encoder:** The state encoder $\phi(f_t, x_0)$ is a transformer-based [52] network that processes all $N$ keypoints in the $f_t$ and the corresponding initial 3D coordinates $x_0$ to output a current tate representation $s_t$, i.e., $s_t = \phi(f_t, x_0)$. For each keypoint, its location $(u_n^t, v_n^t)$ at time $t$ is first encoded using 2D sinusoidal positional encoding and the initial 3D coordinate $x_{0,n}$ is passed through a linear projection head. They are then concatenated together to form a unique descriptor $\epsilon$ for each point. The descriptors for all $N$ points are passed into the state encoder, and we use a CLS token [53] to summarize the current state. Since the points are permutation invariant, we do not add any positional embedding and only add a learnable embedding for the CLS token.

**Temporal Alignment:** During inference, the policy should locate the current task progress and be conditioned on the remaining task flow, rather than the complete task flow, to predict precise actions. To this end, we incorporate a temporal alignment module $\psi$ to predict the remaining task flow given the current keypoints location and the complete task flow. Furthermore, the alignment module enables our method to take in a human demonstration as policy input and learn from it during inference (i.e., demonstration-conditioned execution), a capability that ATM [14] lacks. The idea is similar to the skill alignment in XSkill [45] but we extend it into more complex flow domain.

Instead of predicting the raw remaining task flow, we conduct it in the latent space $\mathcal{Z}$ to improve training efficiency. We construct a transformer-based temporal alignment model $\psi(\mathcal{F}_{0:T}, s_t, \rho_t)$ to predict the latent representation of the remaining task flow $z_t$ based on the complete task flow $\mathcal{F}_{0:T}$, the current state $s_t$, and the robot proprioception $\rho_t$. For the alignment module supervision $\hat{z}_t$, we use another transformer encoder $\xi$ to encode the ground truth remaining task flow $f_{t:T'}$ into the latent space which is accessible in the training dataset $\mathcal{D}$. We use $L_2$ loss between $z_t$ and $\hat{z}_t$, i.e., $\|\hat{z}_t - z_t\|^2$ to supervise the alignment learning.

**Diffusion Action Head:** During inference, the policy can directly condition on $z_t$, i.e., $p(\mathbf{a_t}|z_t, s_t, \rho_t)$ to predict future actions. During training, the policy is conditioned on $\hat{z}_t$ such that the encoder $\xi$ gets supervision. Although the whole system can be trained in an end-to-end fashion, we detach the $\hat{z}_t$ when computing alignment loss, i.e., $\|\hat{z}_t^{\text{detach}} - z_t\|^2$ for more stable training.

## 4 Evaluation

We demonstrate Im2Flow2Act's capability across 4 tasks ranging from rigid, articulate, and deformable objects, including pick-and-place, pouring, open drawer, and folding cloth. The task description can be seen in the leftmost column in Fig. 4.

### 4.1 Experiment Details

**Training data:** We collect *robot exploration* data in simulation and *human hand* demonstration in the realworld. **(i.)** In simulation, we collect data across rigid, articulated, and deformable objects using a UR5e robot through a set of predefined random heuristic actions for exploration. The details for each exploration strategy can be found in the supplementary material. We trained one imitation learning policy with all exploration data. **(ii.)** In real world, we collect *human hand* demonstrations for each task. We mixed the data together to train the flow generation model.

**Comparisons.** We compare our method with the following baselines:1) **ATM:** ATM [14] trains a closed-loop flow generation model that outputs grid flow, and the manipulation policy takes in the immediate next few steps of flow prediction. This comparison tests the benefits of generating complete task object flows when learning from cross-embodiment demonstrations. 2) **Heuristic:** A heuristic action policy selects object contact points and applies a pose estimation method, such as RANSAC, on the future object flow (requiring 3D flow) to infer robot actions, which is implemented in General Flow and Track2Act [15, 17]. We provide the ground truth 3D flows as heuristic method's input. This baseline examines the necessity of having a learning-based policy for translating flow into actions. 3) **Grid Flow:** Similar to ATM, we replace the initial object keypoints in Im2Flow2Act with a set of uniformly sampled keypoints. 4) **No alignment:** To ablate our alignment module design, we removed the alignment model $\psi$ and the policy condition on the complete task flow.

**Evaluation Setting.** We evaluate Im2Flow2Act and the baseline models in two scenarios: (1) demonstration-conditioned execution and (2) language-conditioned execution under both real-world and simulated environment. In the simulations, we utilize a sphere robot to create cross-embodiment demonstrations. In the demonstration-conditioned setting, we use object flow extracted from a single human demonstration video (or a sphere demonstration in simulation) as input for the manipulation policy. This setting evaluates the robustness of the low-level action generation independent of flow generation accuracy. In the language-conditioned execution, we evaluate the full system by first generating flow using learned flow generation network conditioned on the task description and initial frame. The manipulation policy then uses this generated flow for action inference.

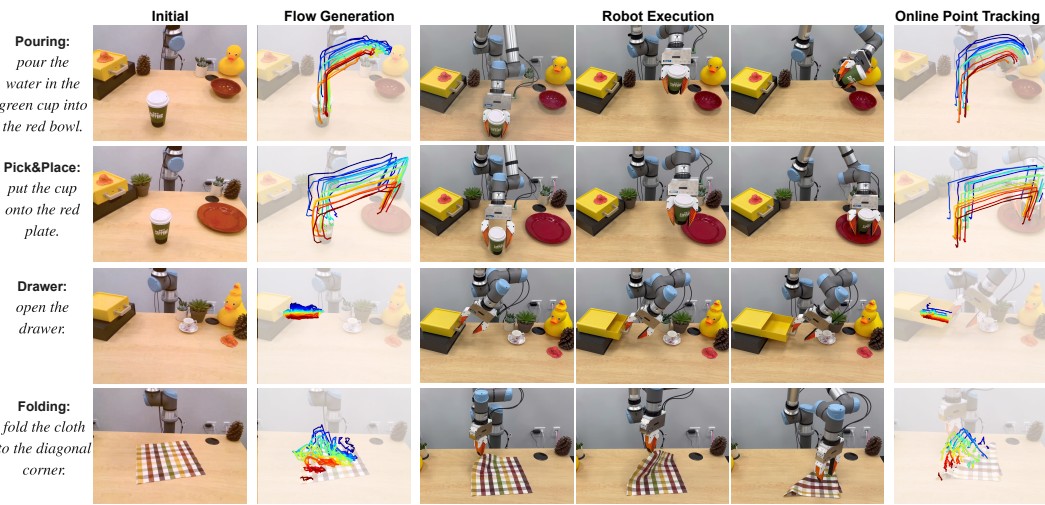

Figure 4: **Task and results:** We conducted evaluations on four tasks involving rigid, articulated, and deformable objects. The initial scene, flow generation visualizations, and online point tracking during inference were captured using the RealSense camera. Visualizations of the robot's execution were recorded from a different angle to provide a clearer view of the robot execution.

Table 1: **Simulation Results (%)**

|  | Demonstration-conditioned execution | | | | Language-conditioned execution | | | |
|---|---|---|---|---|---|---|---|---|
|  | Pick&Place | Pouring | Drawer | Cloth | Pick&Place | Pouring | Drawer | Cloth |
| **Im2Flow2Act** | **100** | **95** | **95** | **90** | **90** | **85** | **90** | 35 |
| **ATM[14]** | / | / | / | / | 50 | 30 | 85 | 30 |
| **Heuristic[15, 17]** | 70 | 50 | 30 | 0 | / | / | / | / |
| **GridFlow** | 30 | 25 | 35 | 45 | / | / | / | / |
| **No alignment** | 80 | 85 | 90 | 90 | / | / | / | / |

## 4.2 Key Findings

**Flows can effectively bridge different data source:** Im2Flow2Act achieves best performance among all baseline in both simulation and realworld, as shown in both Tab 1 and Tab 2. Further, the performance only drops 15% on average in realworld compare to in simulation. As shown in Fig. 4, our learned policy can largely follow the generated flow to complete the desired tasks across rigid, articulated and deformable objects. This suggests that object flow is a good interface to connect the both cross-embodiment and cross-environment data sources.

**Learning-based policy is necessary for translating flow to actions:** To map the generated flow to robot actions, we used an learned policy and alignment network. In contrast, prior works [17, 15] use heuristic-based policy to compute robot action from the flow. When compared to them, we provide the ground truth 3D flows and optimal grasping pose, which are not required by our method. As shown in Tab.1, heuristic-based policy shows good performance for rigid objects, such as pick&place and pouring in simulation. However, it struggles when manipulating deformable and articulated objects in both sim and real as the performance of heuristic methods heavily depends on the camera view. For instance, in the task of opening a drawer, if the center of the point clouds under camera view is not a good contact point, it will push the drawer back (Fig. 5). In the realworld (Tab. 2), we notice the heuristic-based frequently triggers the emergency stop from UR5e as UR5e does not have impedance control, therefore a small error in pose estimation can cause the task to fail. This suggests that a learning-based policy is necessary for translating flow to accurate and safe actions.

Table 2: **Real-World Results (%)**

|  | Demonstration-conditioned execution | | | | Language-conditioned execution | | | |
|---|---|---|---|---|---|---|---|---|
|  | Pick&Place | Pouring | Drawer | Cloth | Pick&Place | Pouring | Drawer | Cloth |
| **Im2Flow2Act** | **95** | **80** | **90** | **70** | **90** | **80** | **85** | 70 |
| **Heuristic[15, 17]** | 70 | 50 | 30 | 0 | / | / | / | / |
| **No alignment** | 55 | 0 | 80 | 60 | / | / | / | / |

**Object-flow is crucial when learning from cross-embodiment demonstration:**. We evaluate the object-centric flow design by comparing our method with ATM. As shown in Tab. 1, Im2Flow2Act outperforms ATM by an average of 30% across four tasks. The key reason ATM performs poorly when learning from cross-embodiment demonstrations is that the visual input becomes out-of-distribution during robot deployment. During the training, ATM's takes in the visual input with cross-embodiment demonstrations (human arm in the real world and sphere in simulation) and makes predictions, most of which are related to the embodiment flow. However, during inference, the visual input contains the robot arm instead of the human arm or sphere. As the model also generates the embodiment flow, the unseen embodiment causes the model to be out-of-distribution. In contrast, Im2Flow2Act only generates the object flow and does not focus on the embodiment. The initial object keypoints also provide a strong condition for the model to generate object-centric flows. Therefore, our model is robust to the presence of different embodiments during robot deployment. We further compare the design decision on initial keypoint sampling by replacing the object-flow in our method with the uniform grid flow. In Tab. 1, we observed that its performance is significantly drop, as the uniform flow also captures motion from the embodiment itself, similar to ATM. This highlights the superiority of object-flow over grid-flow for cross-embodiment learning.

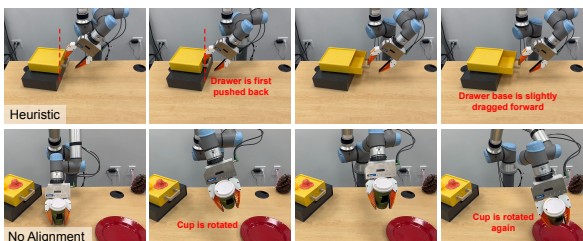

**The alignment module is necessary for leveraging unstructured exploration data:** In Tab. 1, the performance of Im2Flow2Act without alignment only slightly decreased in the simulation. However, we observed that the robot's execution became unsmooth, involving many unnecessary actions such as sudden rotations. This suggests that the policy cannot align its behavior with the task flow based on task progress and only attempts to find the closest trajectory in the exploration data. Further, in realworld, its performance drops significantly (Tab. 2), es-

Figure 5: **Typical failure cases of baselines.** Even given the ground truth flows produced by human, the heuristic policy may produce inaccurate actions, for example, pushing the drawer back first. As for the No Alignment model, the cup may be randomly rotated along the trajectory, which is the same behavior as the random exploration data.

pecially in the pick-and-place and pouring tasks, as the object flow yielded from human can differ from the flow in the simulation. It is challenging for the policy to find similar trajectories in the latent space, leading to random trajectories as shown in Fig. 5. Im2Flow2Act with alignment does not suffer from this issue, as the latent space has finer granularity with the extra alignment supervision, leading to smooth interpolation.

### 4.3 Limitation and Future Work

Our system uses 2D flow as the manipulation interface, enabling us to utilize large-scale 2D videos. However, this approach is inherently ambiguous for representing 3D actions. For instance, a common failure in pouring tasks arises when the robot's z-axis movements (in camera coordinates) lack precision. Additionally, 2D flow struggles with tasks involving out-of-plane rotation (e.g., screwing), a limitation potentially addressed by 3D flow [54, 15]. Our framework also assumes consistent object dynamics between simulation and real world, yet simulating deformable objects remains challenging. Consequently, we observe a significant performance drop in the folding task when deploying the policy in the real world. Please see Appendix I for more discussion.

## 5 Conclusion

We introduce Im2Flow2Act, a scalable learning framework that enables robots to acquire diverse manipulation skills from cost-effective cross-domain data using object flow as a unifying interface. Our final system demonstrates strong real-world manipulation capability by outperforming all baselines across various types of manipulation tasks. Our work paves the way for future research to scale up robotic manipulation skills from diverse data sources.

**Acknowledgments**

We would like to thank Yifan Hou, Zeyi Liu, Huy Ha, Mandi Zhao, Chuer Pan, Xiaomeng Xu, Yihuai Gao, Austin Patel, Haochen shi, John So, Yuwei Guo, Haoyu Xiong, Litian Liang, Dominik Bauer, Samir Yitzhak Gadre for their helpful feedback and fruitful discussions.

Mengda Xu's work is supported by JPMorgan Chase & Co. This paper was prepared for information purposes in part by the Artificial Intelligence Research group of JPMorgan Chase & Co and its affiliates ("JP Morgan"), and is not a product of the Research Department of JP Morgan. JP Morgan makes no representation and warranty whatsoever and disclaims all liability, for the completeness, accuracy or reliability of the information contained herein. This document is not intended as investment research or investment advice, or a recommendation, offer or solicitation for the purchase or sale of any security, financial instrument, financial product or service, or to be used in any way for evaluating the merits of participating in any transaction, and shall not constitute a solicitation under any jurisdiction or to any person, if such solicitation under such jurisdiction or to such person would be unlawful. This work was supported in part by NSF Award #2143601, #2037101, and #2132519. The views and conclusions contained herein are those of the authors and should not be interpreted as necessarily representing the official policies, either expressed or implied, of the sponsors.

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

## Appendix

## A   Additional Related Work

**Sim2Real Transfer.** To overcome the data collection challenge in the real world, a desired approach is to train a robot policy in simulation and deploy it in the real world. Due to the sim-to-real gap, various techniques have been employed to bridge this gap, including but not limited to using depth observation [55, 56, 57, 58, 59], domain randomization [60, 61, 62, 63, 64, 65], knowledge

distillation [66, 67], and system identification [68]. Learning a sim-to-real policy typically requires task-specific environment construction such as reward function design. In contrast, Im2Flow2Act learns the task information from real-world human video by learning a flow generation network. Our policy learn from simulated exploration data completely generating by a set of predefined primitive actions, which is easier to collect and further reduces the need for task-specific environment construction.

# B   Additional Experiment

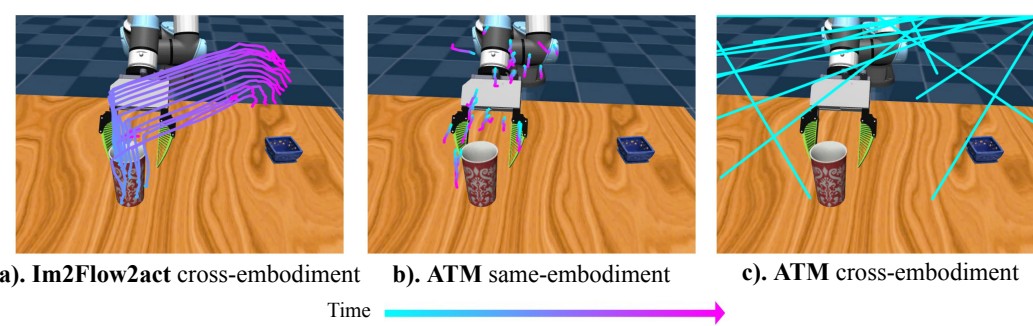

a). **Im2Flow2act** cross-embodiment    b). **ATM** same-embodiment    c). **ATM** cross-embodiment

Time ➤

Figure 6: **Flow generation comparison during robot deployment.** We compare Im2Flow2Act and ATM side by side in the pouring task. Im2Flow2Act (**a**) demonstrates its robustness in the presence of the UR5 in the visual input, generating high-quality object-centric task flows. In contrast, ATM generates noisy flows (**c**) when learning from sphere demonstrations and only performs well (**b**) when learning from same-embodiment demonstrations. Notice, we only visualize the query points with high variance for ATM. As ATM only predicts the future N steps and the mug will not move under the prediction horizon, therefore there is no flows on the mug in sub-figure **b**. Please refer to Fig. 10 for the case robot is more close to the object. We visualize the complete grid flow with object moving under the prediction.

In this section, we provide additional qualitative and quantitative comparisons with the baseline ATM [14] in Pick&Place, Pouring and Drawer task. We used the official ATM repository to train its flow generation model and adapted its manipulation policy to diffusion action head. We remove the RGB input into manipulation policy, similar to Im2Flow2Act. We additionally provide initial depth input same as Im2Flow2Act to ensure a fair comparison. Instead of training on more challenging exploration dataset as in Im2Flow2Act, we directly provide task specific data to train ATM manipulation policy. For ATM, we trained two versions of its flow generation model: one with UR5 demonstrations (i.e., same embodiment) and one with sphere agent demonstrations (i.e., cross-embodiment demonstrations). Note that our method is always trained with cross-embodiment demonstrations.

|  | Demonstration-conditioned execution | | | Language-conditioned execution | | |
|---|---|---|---|---|---|---|
|  | Pick&Place | Pouring | Drawer | Pick&Place | Pouring | Drawer |
| **Im2Flow2Act** | **100** | **95** | **95** | **90** | **85** | **90** |
| **ATM cross embodiment** | / | / | / | 50 | 30 | 85 |
| **ATM same embodiment** | / | / | / | 90 | 90 | 95 |

Table 3: **Comparison with ATM Result (%)**

In Fig. 6, we first compare the generated flows from both Im2Flow2Act and ATM side by side during robot deployment. As shown in Fig. 6.a, Im2Flow2Act is robust to the presence of the UR5 in the visual input and generates high-quality flows, thanks to the choice of using complete object-centric flow. In contrast, ATM can only generate high-quality flow (see Fig. 6.b) when trained with the same embodiment (i.e., trained with UR5 and inferred with UR5). It generates low-quality and

noisy flows (see Fig. 6.c) when trained with cross-embodiment demonstrations (i.e., trained with a sphere agent and inferred with UR5).

As mentioned in updated main paper Sec. 3.1., the key reason ATM performs poorly when learning from cross-embodiment demonstrations is that the visual input becomes out-of-distribution during robot deployment. Compared to Im2Flow2Act's object-centric flow, ATM generates grid flow. This design choice leads to the majority of the generated flow is for the embodiment. During the training, ATM's flow generation model takes in the visual input with cross-embodiment demonstrations (human arm in the real world and sphere in simulation) and makes predictions, most of which are related to the embodiment flow. However, during robot deployment, the visual input contains the robot arm and gripper instead of the human arm or sphere. As the model also generates the embodiment flow, the unseen embodiment causes the model to be out-of-distribution.

In contrast, Im2Flow2Act only generates the object flow and does not focus on the embodiment. The initial object keypoints also provide a strong initial condition for the model to generate object-centric flows. Therefore, our model is robust to the presence of different embodiments during robot deployment.

We further evaluate the complete system of ATM with the two versions of the flow generation model and compare them with Im2Flow2Act. Notice, ATM system lacks the ability to condition on complete demonstration. As shown in Tab. 3, ATM's performance significantly drops when the flow generation model is learned from cross-embodiment demonstrations compared to when it is learned from same-embodiment demonstrations. In contrast, our system demonstrates strong capability when learning from cross-embodiment demonstrations.

## B.1 Long Horizon Task with Multiple Objects

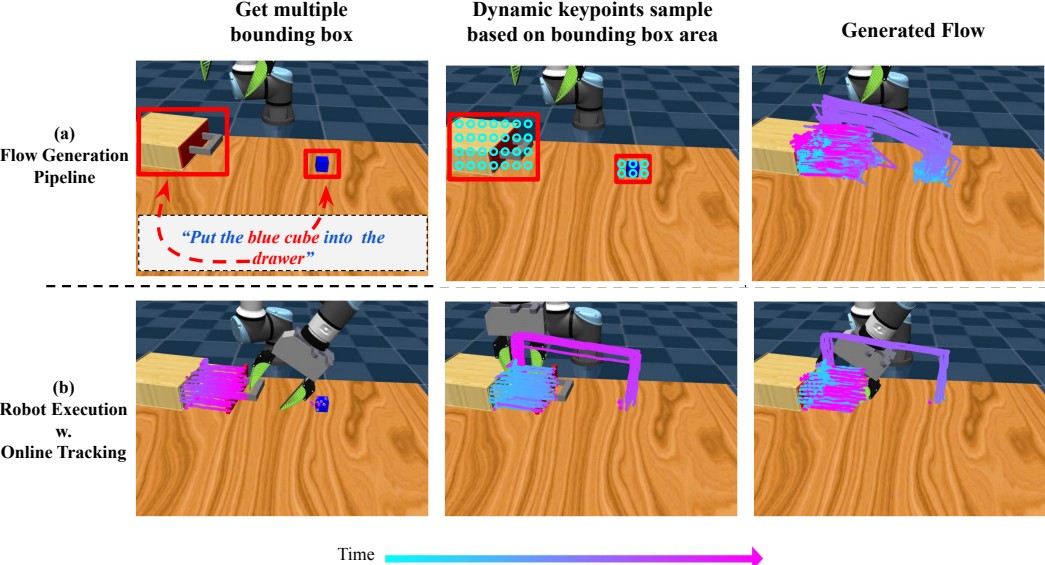

Figure 7: **Long-Horizon Task with multi-objects:** Im2Flow2Act is capable of generating flows (**a**) for long-horizon tasks involving multiple objects. Our pipeline starts with getting multiple bounding box for objects of interest and dynamically sampling the initial keypoints proportional to the bounding box area. The sampled keypoints are then concatenated to form a $H \times W$ rectangular flow image as the flow generation model condition. Our manipulation policy (**b**) effectively follows the generated flow to successfully complete the tasks. For videos, please visit our anonymous website: https://im2flow2act.github.io/

We conducted an additional experiment to evaluate Im2Flow2Act's performance on a long-horizon task involving multiple objects. In this task, the robot needs to perform a sequence of actions: first, open the drawer, then pick up the blue cube from the table and place it into the drawer, and finally

close the drawer. The positions of the drawer and the cube are randomly initialized at the beginning of each episode. We collected 150 cross-embodiment demonstration trajectories (i.e., using a sphere agent) and trained our flow generation model accordingly and 100 UR5 demonstration to train the manipulation policy.

For tasks involving flow generation for multiple objects, we similarly use Grounding DINO to obtain the bounding boxes of all relevant objects. We then perform uniform sampling within each bounding box. The number of keypoints sampled within each bounding box is proportional to its area. To ensure a total of $H \times W$ keypoints, we pad the final sampled keypoints as needed. The key idea is still to form structure flow image from permutation invariant keypoints.

The key reason Im2Flow2Act is capable of generating long-horizon flow is that our flow generation model generates flow with temporal sub-sampling, rather than at the original flow speed. This design make it easy for the network to generate a compact flow for long-horizon tasks. Additionally, it makes the system more robust when the speed of cross-embodiment demonstrations differs significantly from the robot's speed.

The same parameters were used for training and inference for both flow generation network and manipulation policy as in the other tasks, with further details provided in appendix Sec. F.1 and Sec. F.2

| | Demonstration-conditioned execution | Language-conditioned execution |
|---|---|---|
| **Im2Flow2Act** | 90 | 85 |
| **ATM** | / | 45 |

Table 4: **Long Horizon Task with Multiple Objects Result (%)**

We compared Im2Flow2Act's performance with ATM using the same training data for flow generation and manipulation policy as Im2Flow2Act. As shown in Tab. 4, Im2Flow2Act successfully completes long-horizon tasks with 90% success rate in the demonstration-conditioned execution setting and 85% success rate in the language-conditioned execution setting. In the language-conditioned execution setting, we visualized the generated flow and the robot execution in Fig. 7. As shown in the Fig. 7, Im2Flow2Act is capable of generating complete object-centric task flows for long-horizon tasks with multple objects, and the robot execution closely aligns with the generated flows. ATM performance suffers from the out-of-distribution visual input into the flow generation model as we discussed in above section and our updated main paper Sec. 3.1.

## B.2 Ablation Study on Initial 3D Keypoints

| | Demonstration-conditioned execution | | | Language-conditioned execution | | |
|---|---|---|---|---|---|---|
| | Pick&Place | Pouring | Drawer | Pick&Place | Pouring | Drawer |
| **Im2Flow2Act** | 100 | 95 | 95 | 90 | 85 | 90 |
| **Im2Flow2Act w.o. 3D** | 100 | 90 | 90 | 85 | 85 | 80 |

Table 5: **Ablation Study on Im2Flow2Act w./w.o initial 3D keypoints Result (%)**

We conduct an additional experiment to demonstrate the necessity of having initial 3D keypoints. Our experiments (see Tab. 5) show that the initial 3D keypoints improve policy performance when taking generated flow as input (i.e., language-conditioned execution). The generated flow often contains some noisy keypoint movement, especially for keypoints around objects. The additional 3D keypoints help the network distinguish between noisy keypoints and those actually on the objects, e.g., drawer handle.

Once the initial 3D keypoints have been identified, all remaining movement is relative to these initial 3D keypoints. Therefore, our system does not require 3D keypoints for all frames. Moreover, obtaining 3D keypoints for every frame in the real world involves additional operations, so we aim to keep our system simple and efficient.

## B.3 Generated Flow in Simulation

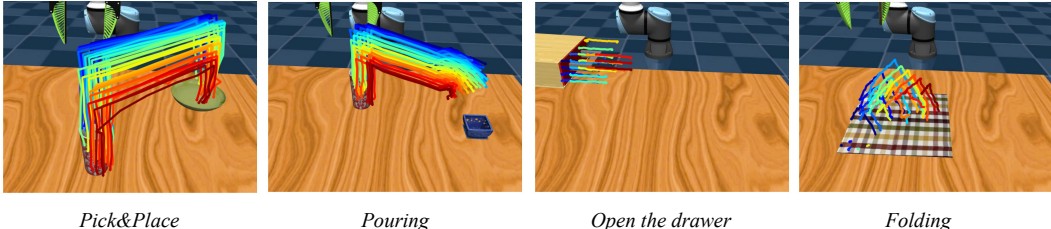

| *Pick&Place* | *Pouring* | *Open the drawer* | *Folding* |

Figure 8: **Generated Flow in simulation:** We visualize the generated flow with high-variance for four tasks in simulations. We use the motion filter to filter the flow with high-variance. More details for motion filters can be found in appendix G.2. The keypoints has been down-sampled to provide better visualization. Note that in the real world, we train the flow generation model using human demonstrations.

In the simulation, we collect cross-embodiment demonstrations by replacing the UR5 with a sphere agent and use these demonstrations to train the flow generation model. The learned flow generation model is used to evaluate Im2Flow2Act's performance in simulation under the language-conditioned execution setting (see Tab. 1).

## B.4 Ablation on pretrain Stable Diffusion

The quantitative results of whether to use a pre-trained Stable Diffusion model to train the flow generation model are presented in Appendix Sec. D. In this section, we visualize the training loss for both approaches. As shown in Fig. 9, leveraging the pre-trained U-Net from Stable Diffusion leads to faster convergence and lower loss. This suggests that fine-tuning on a pre-trained U-Net is beneficial for flow generation.

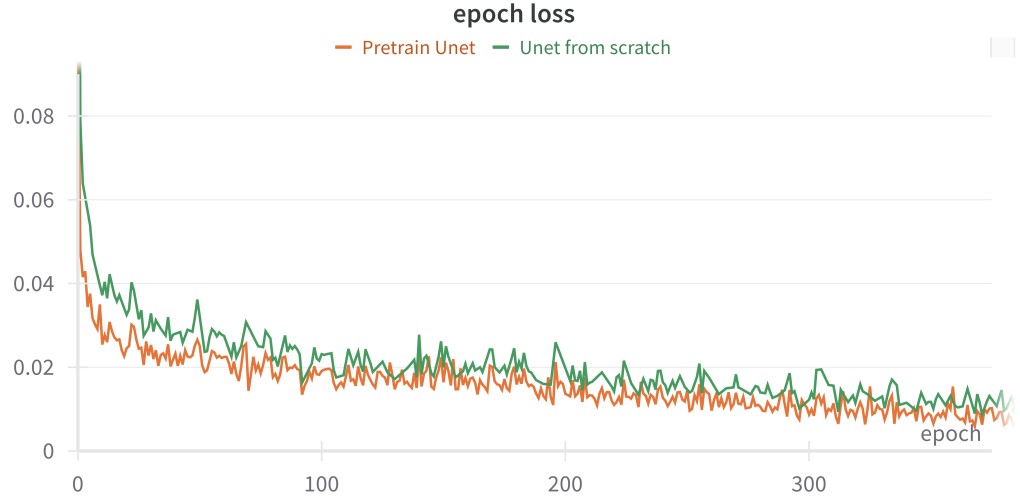

Figure 9: **Unet Pretrain vs Scratch:** The training loss converges much faster for pretrain Unet. The visualized loss is not being smoothed.

## B.5 ATM flow generation additional visualization

In this section, we provide the visualization for complete grid flow prediction from ATM and also the filleted object flow from predicted grid flow in Fig. 10. The ATM predicted the grid flow for the future 16 steps based on the current state. The bounding box we use to to filter the object flow is the same as the one we used for Im2Flow2Act. Compared to Fig. 6, the robot is more close to the object in Fig. 10. From Fig. 10 **b1** and **b2**, the ATM can generate high-quality flows if learning from the same embodiment demonstrations. However, the generated flow is completely noise if learning from the cross-embodiment demonstration (See Fig. 10 **c1** and **c2**). In contrast, Im2Flow2Act can generate high-quality flow even when learning from cross-embodiment demonstrations. Notice, we increased the number of grid key points from 32 in the original ATM paper to 400 to ensure fair comparison. The ATM policy also takes in the complete flows from 400 grid keypoints as input, which is significantly larger than 128 object key points in our method.

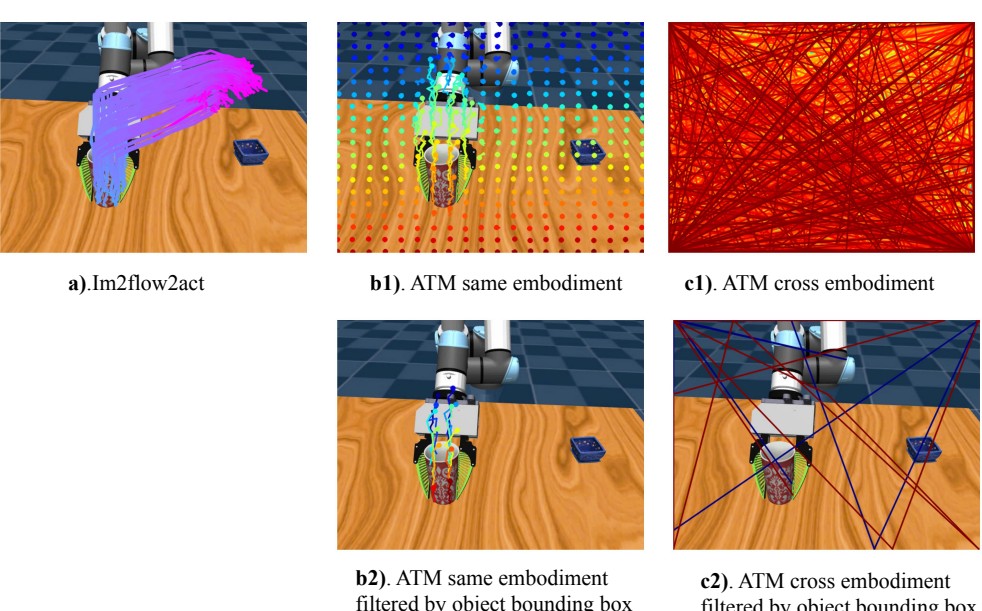

**a)**.Im2flow2act          **b1)**. ATM same embodiment          **c1)**. ATM cross embodiment

**b2)**. ATM same embodiment filtered by object bounding box          **c2)**. ATM cross embodiment filtered by object bounding box

Figure 10: **Flow generation comparison during robot deployment (grid flow and filterd with object bounding box)** We compare Im2Flow2Act and ATM side by side in the pouring task. ATM generates noisy flow when learning from cross-embodiment demonstrations (**c1,c2**). In contrast, Im2Flow2Act generated high-quality flow even when learning from cross-embodiment demonstrations (**a**). Notice, the visualized flow from Im2Flow2Act is after processed by motion filters. The initial keypoints is uniformly sampled from the bounding box and we do not manually select the initial points.

## C    Data Collection

In Im2Flow2Act, we collect two types of data: a simulated robot exploration dataset and real-world human demonstration videos. Note that we do not collect any real-world robot data. Below, we detail the data collection process.

### C.1    Simulated exploration Data

We use MuJoCo as our simulation engine and have constructed three types of exploration environments featuring rigid, articulated, and deformable objects. An UR5e robot explores these environments using a set of predefined random heuristic actions.

**Rigid:** In the rigid environment, the robot can interact with five different objects, including a toy car, a shoe, a mini-lamp, a frying pan, and a mug. At the beginning of each episode, one object is randomly selected and placed on the table. The robot first picks up the object and starts executing 6DoF random trajectories. A random trajectory is constructed as a 3D cubic Bézier curve with four key waypoints: the start point $p_0$, two control points $p_1$ and $p_2$, and the end point $p_3$. The start point is where the robot first picks up the object, and the end point is randomly selected within the robot's reachable world coordinates. The two control points are obtained by adding curvature to the line defined by $p_0$ and $p_3$. Specifically, for the first control point $p_1$, we first calculate the one-third point $m_1$ between $p_0$ and $p_3$, i.e., $m_1 = p_0 + \frac{(p_3 - p_0)}{3}$. We then twist $m_1$ in 3D space to obtain the first control point by adding a random offset (curvature), i.e., $p_1 = m_1 + \epsilon$, where $\epsilon \in \mathbb{R}^3$. We set each dimension of $\epsilon$ be uniformly sampled from $(-0.05, 0.05)$. Similarly, we obtain the second control point by calculating the two-thirds point and adding some random offset. Once we obtain the points $p_0$, $p_1$, $p_2$, and $p_3$, we define the cubic Bézier curve by $(1 - t)^3 P_0 + 3(1 - t)^2 t P_1 + 3(1 - t)t^2 P_2 + t^3 P_3$, where $t \in [0, 1]$. We equally sample $k$ ($k = 16$) waypoints along this curve, which defines the translation for the robot's exploration trajectory. We add a target object orientation (random sample from $(-\frac{\pi}{8}, \frac{\pi}{8})$ for each dimension) when the object arrives at the final point $p_3$. We interpolate between the initial orientation at $p_0$ and the target orientation at $p_3$ and attach them to the $k$ waypoints. For each episode, we sample two consecutive 3D cubic Bézier curves, allowing the robot to interact with random objects and build the correspondence between flows and actions. The trajectory ends with a random rotation or place actions.

**Articulated:** We construct various types of drawers, none of which are the same as those tested in the real world. In each episode, one drawer is selected and placed on the table. The robot explores these articulated objects by opening the drawer to different extents, not necessarily fully open. To open a drawer, a random contact point is selected on the drawer handle within a range of $(-0.07, 0.07)$ from the center of the handle. This contact point is then passed to our predefined opening primitive. The primitive places the gripper on the contact point and initiates the drawer's opening by pulling perpendicular to the drawer's surface. A random threshold, ranging from $(0.07, 0.12)$, is sampled to define the success criterion for each episode. The robot stops execution once the drawer handle has moved beyond this threshold, indicating a successful opening.

**Deformable:** At the beginning of the episode, a cloth is randomly placed on the table. The robot has the option to grasp either the left or right corner of the cloth. We use the predefined grasp primitives to achieve this behavior, which receive the grasping pose as input, similar to the one we use in the rigid dataset generation. Once grasped, the robot begins random folding (not necessarily folding diagonally). We have defined nine different folding targets, and the robot randomly selects one to start folding. These nine different folding targets is formed by the combination of $(0.08, 0.15, 0.28)$ on x and y axis of the grasping point coordinate. During this process, we also vary the folding trajectory by lifting the cloth to different heights. We use the predefined move heuristic to move the end effector to a random height. We first calculate the distance $d$ between the folding corner and the folding targets. We then randomly sample the lifting height between $(d/4, d)$. Finally, we move end effector towards to the folding target and open the gripper.

In total, we collect 4800 random exploration trajectories. Once all data is collected, we use an iterative procedure to obtain object flows. For each collected episode, we begin by sampling uniform grid (30x30) keypoints on the initial frame and track all points using Tapir [48], similar to the approach in ATM [14]. We then apply moving filters and SAM (Segment Anything Model) filters which we will explain in details at section G below to obtain keypoints on the object. However, as the initial sampling is uniform, there may be few points located on the objects. To address this, we sample additional points around the existing keypoints based on the SAM output to achieve a denser distribution of object keypoints. Specifically, for segments containing keypoints after filtering, we sample proportionally based on the number of filtered keypoints in that segment. In total, we sample 900 points on each object. Finally, we run the point tracking algorithm again on the newly sampled object keypoints to obtain dense object flow.

### C.2 Real-world Human Demonstration Video:

We collect in-domain human demonstration videos for four tasks: pick & place, pouring, opening drawers, and folding cloth. We use a RealSense camera to record the demonstrations at 30 FPS. The descriptions below also serve as detailed task descriptions. In Im2Flow2Act, we use the human videos to provide the system with task information. We define the robot base as the origin of the world coordinate system and use the Right-Handed Coordinate System.

- **Pick & Place:** Pick up a green cup and place it into a red plate. The red plate is randomly placed on one side of the table, while the cup is placed randomly in the middle of the table.
- **Pouring:** Pick up a green cup and hover it over a red bowl, then rotate the cup towards the bowl. The initial placement of the cup and the bowl is the same as in the pick & place task.
- **Drawer Opening:** Fully open a drawer that is randomly placed on one side of the table. The pose of the drawer, including its position and orientation, can vary. The drawer is not rigidly attached to the table.
- **Cloth Folding:** Fold a 33 cm x 33 cm cloth from one corner (either lower left or lower right) to the diagonal. The cloth is randomly placed in the middle of the table.

Once the human demonstration is collected, for each episode, we use Grounding DINO to obtain the object bounding box at the initial frame and uniformly sample the keypoints inside, where we set $H = W = 32$ as sampling parameters. We then run point tracking algorithm to track the keypoints across frames in the episode. To construct the training dataset, we uniformly sample 32 frames from each episode to form the task flow.

## D  Ablation study

We conducted an ablation study in simulation to evaluate the impact of using pretrained StableD-iffusion (SD) versus training it from scratch on the flow generation model. In this ablation study, we still use pretrained AE (auto-encoder) from the SD but trained the U-Net from scratch instead of incorporating LoRA layers. To ensure a fair comparison, we deploy the same flow-conditioned

Table 6: **Ablation study Results (%)**

|  | Pick&Place | Pour | Drawer | Cloth |
|---|---|---|---|---|
| Pretrain U-Net | 90 | 85 | 90 | 35 |
| U-Net From Scratch | 90 | 90 | 95 | 30 |

policy for both the pretrained SD and the training scratch as the manipulation policy. We collect data for four tasks in the simulation by substituting the UR5e robot with a sphere robot to create a cross-embodiment scenario. Both networks were trained for the same number of epochs and evaluated within the same initial state. Based on the results shown in Tab. 6, the choice of pretraining had a minor impact on Im2Flow2Act's final performance. This suggests that: *i.* Although StableDiffusion was initially designed for image generation, directly using its pretrained weights for flow generation does not impact its performance. Utilizing LoRA can lead to better training efficiency compared to training from scratch. *ii.* The latent space encoded by the AE from the pretrained SD might benefits the diffusion model's learning process.

Compared to the results using ground truth flow in simulation, we observed that for tasks like pick & place, pouring, and drawer opening, the success rates by taking generated flow as input are very close. This further demonstrates flow generation network's capabilities. However, the success rate for cloth folding drops significantly. We attribute this to the way we constructed the cloth folding simulation environment. Grasping deformable objects like cloth is challenging, so we attached a cube (See Fig. 11) at the corner of the cloth to help the robot to lift the corner by grasping the cube. During data generation and inference, we render only the cloth, not the cube, to mimic realistic cloth folding behavior. We made the cube's dimensions on the xy-plane very small (1cm x 1cm) to emulate the width of a gripper grasping actual cloth. Since the output from a diffusion

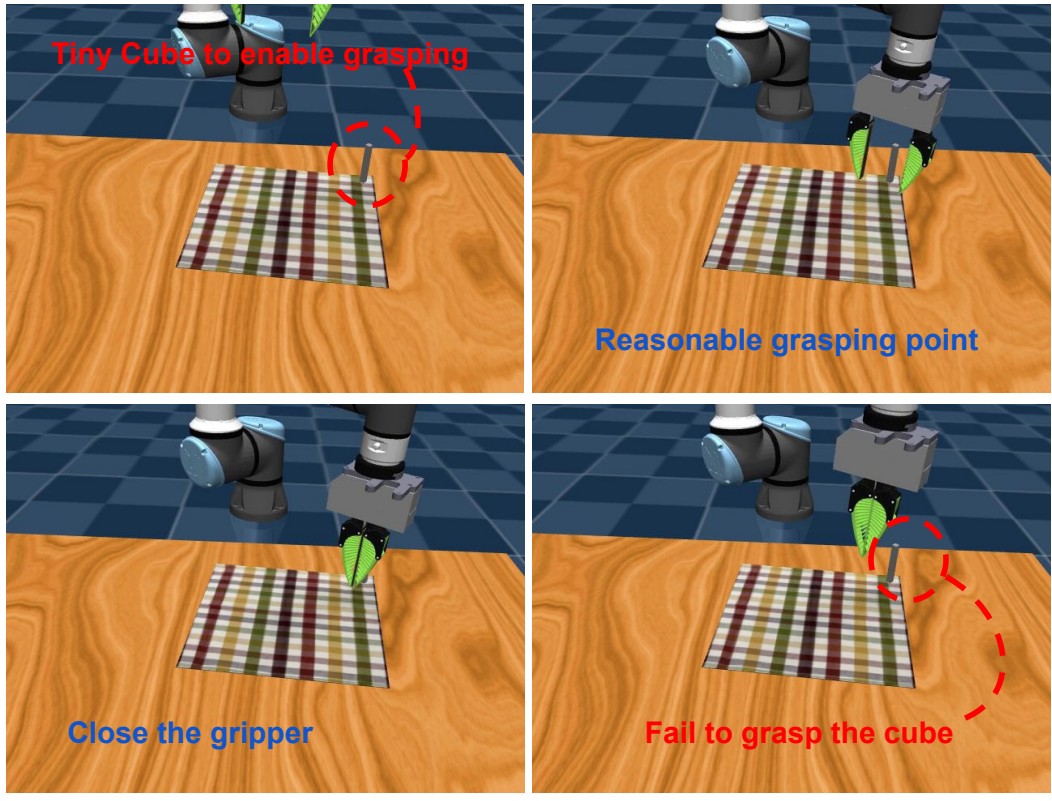

Figure 11: **Policy rollout in Deformable Environment.** We attach a tiny cube (1cm x 1cm x 9cm) to enable robot grasp deformable objects. The generated flow guides the policy toward an appropriate grasping point; however, it fails to grasp the tiny cube, resulting in task failure.

model typically exhibits multimodal distributions and our training dataset contains considerable randomness, the learned flow generation model produces reasonable flows for the folding task, but may not direct the policy to grasp the cube precisely. Moreover, the flows after motion filtering exacerbate the issue. We found that in most cases, the robot can reach the corner and attempt the grasp (See Fig. 11), but it fails to accurately grasp the cube. In contrast, we observed a reasonable success rate in real-world experiments because, in practice, the robot can grasp the cloth anywhere along the corner to executing folding.

# E    Experimental Details

## E.1    Real-World Setup

We use a UR5e robot equipped with a WSG-50 gripper. During inference, the UR5e receives end-effector space positional commands from a 2.5 Hz policy. We limit the end-effector's speed to less than 0.2 m/s and restrict its position to at least 1 cm above the table for safe execution. A RealSense D415 depth camera is mounted at the table to capture policy observations, including the initial depth image for the initial point cloud $x_0$ and the real-time RGB images used for online point tracking. The RealSense camera records at 720p and 30 Hz. We downsample the resolution to 256x256 at 5 Hz for the online point tracking algorithm to process. Additionally, we use a smartphone positioned at a different angle to better record robot execution. A desktop with a 24 GB NVIDIA RTX 4090 runs the flow generation network and flow-conditioned imitation learning policy inference and online point tracking algorithm.

## E.2 Real-World Evaluation Protocol

In this section, we describe the details of the evaluation protocol in the real world.

*1)* **Initial State:** When evaluating our system, we do not match the background scene setup to those recorded in the human demonstrations such that we can also test the generalization capability of flow generation network. For the initial position of objects, they are placed with roughly the same distribution as in the human video demonstrations.

*2)* **Success Metric:** Below are the details of the real-world success metric for all four tasks:

- **Pick & Place:** The robot needs to pick up the cup and place it into the red plate. We do not require the mug to be centered on the plate. We consider one episode successful if: *i.* the robot can steadily place the mug onto the red plate; *ii.* the plate does not move more than 5 cm on the table during the robot's manipulation process.
- **Pouring:** The robot needs to pick up the cup, hover it near the bowl, and execute pouring actions. We consider one episode successful if: *i.* the robot demonstrates the complete pouring behavior by rotating the cup more than 30 degrees; *ii.* at least half of the cup overlaps with the red bowl on the x-axis in terms of world coordinates.
- **Drawer Opening:** The robot needs to fully open the drawer without colliding with its surface. Since the drawer is not rigidly attached to the table, it may slightly slide during the robot's execution. An episode is considered a success if: *i.* the robot gripper does not hevaily collide with the drawer surface and push the drawer back more than 5 cm; *ii.* the drawer is fully opened;
- **Cloth Folding:** The robot needs to fold the cloth from one corner to the diagonal opposite corner. We consider an episode successful if: *i.* the robot folds the corner as indicated by the generated flow; *ii.* the two corners are within a threshold once the robot completes the execution. Due to the large sim2real gap for deformable simulation, we set this threshold as 7 cm.

## E.3 Evaluation Procedure

**Im2Flow2Act with/without alignment:** For each task, we first record the initial frame with different initial states for 20 evaluation episodes. We then query Grounding DINO on the object of interest to obtain the bounding boxes in all initial frames. Using the bounding box, initial frame, and the task description, we generate the object flow (task flow) for all episodes and store them as a buffer on the disk. With all ingredients for policy inference set, we start policy evaluation for each episode by manually matching the initial states to be close to pixel-perfect within the mounted RealSense camera and load the corresponding generated flow from the previously saved flow buffer.

**Heuristic-based policy:** To obtain ground truth future point clouds for the objects, we first record human demonstrations for 20 evaluation episodes. We store both RGB and depth images during this process. For each evaluation episode, we manually match the initial states to be close to pixel-perfect and obtain the open-loop action sequence by estimating object pose transformations between the initial frame and future frames for each time step in human demonstrations. We use the same motion filters in Im2Flow2Act to ensure fair evaluation. Furthermore, we provide the maximum available points (without downsampling) from the task flow. We manually check the transformed point cloud by overlapping the transformed initial frame point cloud and future frame point cloud to ensure the transformation is largely correct under the noisy conditions of the real-world depth camera.

# F Training Details

## F.1 Flow Generation Network

**Condition Injection.** The text condition are passed into the CLIP [69] text encoder to obtain text embedding. We also input the initial frame and the initial object keypoints $\mathcal{F}_0$ into the model to ensure the generation process considers the spatial relationship between objects and the scene. We use the CLIP image encoder to encode the initial frame, yielding a $P^2$ embedding for each patch from

the last ViT [70] layer. The initial keypoints $\mathcal{F}_0$ are encoded using fixed 2D sinusoidal positional encoding. All conditions are injected into the denoising process through cross-attention.

**Training Details.** For the rectangular flow image, we set the spatial resolution to $H = W = 32$ and $T = 32$, generating flow for 1024 keypoints over 32 steps. We finetune the decoder from StableDiffusion for 400 epochs with a learning rate of $5e-5$. To obtain these keypoints, we uniformly sample them from the bounding box provided by Grounding DINO. For training AnimateDiff, we insert the LoRA (Low-Rank Adaptation) with a rank of 128 into the Unet from StableDiffusion and train the motion module layer from scratch with learning rate of $1 \times 10^{-4}$ for 4000 epochs using AdamW [71] optimizer with weight deacy $1 \times 10^{-2}$, betas $(0.9, 0.999)$ and epsilon $1 \times 10^{-8}$. We load the pretrained (openai/clip-vit-large-patch14) weights from CLIP [69] to process the initial frame and freeze them during the entire training. Zero-initialized linear layers are used to process the patch embedding and the initial keypoints embedding before passing the conditions into the cross-attention layers.

## F.2 Flow-Conditioned Imitation Learning Policy

**Training Data Format:** A training sample consists of $(\rho_t, f_t, \mathbf{a_t}, F_{0:T})$, where $\rho_t$ is the proprioception data, $\mathbf{a_t}$ is a sequence of actions $a_t, \ldots, a_{t+L}$ of length $L$, and $f_t$ contains the locations $(u, v)$ of $N = 128$ object keypoints in the image space at time $t$. We set the object flow (i.e., task flow) horizon $T = 32$, which matches the output of the flow generation network. The action sequence length is set to 16. The $N$ keypoints are randomly selected from all available keypoints for every training sample during the training process. To construct the task flow $F_{0:T}$, we randomly select $T$ frames from the episode length $T'$ to which the training samples belong. This allows the policy can be trained with diverse task flow as the flow generation model is trained with human data which typically has different execution pace compared to robot. Further, we constrain the length of task flow $F_{0:T}$ be much shorter than the episode length $T'$ to ease the training difficulty of flow generation model otherwise it needs to generate long-horizon task flows. To ensure the task flows are complete, we include both the first and last frames of the episode in the task flows.

**State Encoder:** We project the keypoints' initial 3D coordinates, $X_0$, into a 192-dimension vector using a linear layer. We also encode keypoints' locations $(u, v)$ in image space into another 192-dimension vector, using a fixed 2D sinusoidal positioning. These two vectors are concatenated to form the descriptor $\epsilon$, with a total dimension of 384. We then pass all keypoints' descriptors into the state encoder $\phi$, which is a transformer with 4 encoder layers. It outputs a state representation of dimension 384 using a CLS token.

**Temporal Alignment:** As discussed in the main paper, during training, we first encode the remaining task flow $f_{t:T'}$ into $z_t \in \mathcal{Z}$. This process involves encoding the keypoints at each time step $f_{t:T'}$ into $s_{t:T}$ via the state encoder. Next, we encode the future state representation $s_{t:T}$ into the latent space through the encoder $\xi$, which is implemented as a transformer with 4 encoder layers. We use fixed 1D sinusoidal positional encoding to preserve temporal information in the state representation $s_{t:T}$ before feeding them into $\xi$. The Temporal Alignment model is implemented as a transformer with 8 encoder layers. We also add fixed 1D sinusoidal positional encoding to all inputs and utilize a CLS token for making predictions.

**Diffusion Action head:** We use the diffusion policy [51] as our action head. We use DDIM scheduler with 50 training diffusion steps and 16 inference steps.

We train the policy for 500 epochs with learning rate 1e-4 using AdamW with weight deacy $1 \times 10^{-2}$, betas $(0.9, 0.999)$ and epsilon $1 \times 10^{-8}$.

# G Inference Details

In this section, we describe the details of the inference process, which includes Grounding DINO, motion filters, and online point tracking.

### G.1 Grounding DINO

For each task, we begin by using Grounding DINO to identify the object of interest. We manually provide the keyword to the model; however, this process could potentially be automated using a large language model to find the desired object in the task description. Specifically, we employ the grounding-dino-base model to extract the object's bounding box. The keywords used for the pick & place and pouring tasks are "green cup". For drawer opening, the keyword is "yellow drawer", and for cloth folding, it is "checker cloth". The input images are processed at a resolution of 480x640.

### G.2 Motion Filters

We use motion filters to process the object flow (i.e., task flow) generated from the flow generation model. As explained in the main paper, the initial keypoints are constructed by uniformly sampling within the bounding box. This approach inevitably yields keypoints that are not on the object, specifically, keypoints that fall on the background. To address this, we deploy several filters simultaneously to remove these background keypoints. Additionally, we implement depth filters to eliminate keypoints that lack depth data from noisy real-world depth image.

**Moving Filter:** In the training set, keypoints sampled on the background remain static in the image space, as only the object is moving. Therefore, we deploy a moving filter during inference time to remove keypoints whose movement in the image space (256x256) is below a certain threshold. We find that this filter effectively eliminates most background keypoints. In real-world experiments, we set the threshold as 20 for pick & place, pouring, and drawer opening tasks, and as 10 for cloth folding.

**SAM Filter:** To further remove points after applying the moving filter, we deploy the Segment Anything Model (SAM) [72]. Specifically, we first resize the initial frame to 256x256 and pass it through SAM to obtain the finest segmentation. We then iterate through the keypoints and filter out those where the area of the located segment exceeds a threshold. We use a high threshold value of 10,000 for all tasks to prevent filtering out keypoints on objects with rich textures.

**Depth Filters:** Real-world depth images are often noisy and contain many "holes." We filter out keypoints where the depth value is missing (i.e., the value is zero).

We randomly select N=128 keypoints which is the same number we used for training after applying motion filters as the policy input.

### G.3 Online Point Tracking:

We utilize the online point tracking function from Tapir [48] to track the filtered keypoints during inference. We resize the visual observations to 256x256 and run the online point tracking at 5Hz.

## H    Simulation Cross Embodiment Demonstration Details

As we described in the main paper, we collect sphere agent demonstration to mimic the human demonstration in the real-world. In this section, we provide details for collection process. We use the same set of pre-defined heuristic action primitive to collect the sphere demonstration. Please refer to the appendix Sec. C for details.

**PickNplace:** At the start of each episode, both the mug and the target plate are randomly positioned on the table. The mug's initial position is randomly sampled along the x-axis between $(0.42, 0.76)$ and along the y-axis between $(-0.1, 0.02)$. The target plate's initial position is randomly sampled along the x-axis between $(0.42, 0.65)$ and along the y-axis between $(0.24, 0.32)$. The robot first executes the grasp primitive, lifting the mug to a randomly selected height between $(0.15, 0.20)$ meters above the table. After lifting the mug, the robot performs the move primitive, positioning the mug above the target plate. The hover position is randomly sampled from $(0.1, 0.25)$. Finally, the robot places the mug onto the plate and opens the gripper.

**Pouring:** The initial position distribution for the mug is the same as in the PickNplace task. The initial distribution for the target bowl is the same as that for the target plate in the PickNplace task. The robot follows the same procedure as in PickNplace before moving the mug, including the primitives used and the random parameters sampled. After lifting the mug, the robot moves it near the target bowl and hovers the mug at a position sharing the same x-axis value. The robot then executes the pouring primitive, which rotates the mug clockwise around the y-axis. The rotation angle is randomly sampled from $\left(\frac{5\pi}{16}, \frac{7\pi}{16}\right)$.

**Drawer:** The drawer's initial pose is randomly sampled at the beginning of each episode. Its initial position is randomly sampled from $(0.45, 0.62)$ on the x-axis, $(-0.4, -0.32)$ on the y-axis, and $(0.05, 0.18)$ on the z-axis. The drawer's initial rotation is sampled from $\left(\frac{3\pi}{8}, \frac{\pi}{2}\right)$. The robot executes the open primitive as described in Appendix Sec. C.

**Folding:** The cloth is randomly placed on the table at the beginning of each episode. We describe the initial position using the corner pinned with a cube. The corner is randomly placed between $(0.32, 0.44)$ on the x-axis and $(-0.32, 0.32)$ on the y-axis. Values beyond $0.44$ on the x-axis often lead to unstable simulations due to IK errors. The cloth is randomly rotated between $\left(-\frac{\pi}{8}, \frac{\pi}{8}\right)$ Instead of randomly sampling from the nine predefined folding points as described in Appendix Sec. C, the robot will only fold the cloth diagonally.

**Long Horizon:** The initial position distribution for the drawer is the same as in the Drawer task. However, the sample range on the z-axis is modified to $(0.05, 0.12)$ to ensure the cube remains within the camera frame when placed into the drawer. The cube's initial position distribution is the same as the mug's in the PickNplace task. The robot executes the open primitive to open the drawer, the grasp primitive to grasp the cube, and finally, places the cube into the drawer. The input parameters for all primitives are the same as those used in PickNplace, with a modified sample range for the hover position before placing, $(0.1, 0.15)$, to keep the cube within the camera frame.

# I  Additional Limitation

The system assumes the camera viewpoint is calibrated between simulation and testing environments, with actions visible from the camera. Moreover, using flow as an action abstraction overlooks some action details, limiting performance in dexterous tasks like in-hand manipulation. Our flow generation model relies on accurate object detection, which could be improved with advanced object detection/segmentation models [73]. Finally, we focus on "manipulation tasks" rather than locomotion, where manipulation is defined by changing object states through interaction. However, in scenarios where object state changes are not reflected in object motion (e.g., "touching a button on a touch screen"), our method may not apply.

