# OpenReview forum: "Flow as the Cross-domain Manipulation Interface"
_robot-learning.org/CoRL/2024/Conference — CoRL 2024_

### Official Review · Reviewer_GbPT · 2024-06-28
**See the review part.**

**Originality:** 2
**Technical Quality:** 2
**Clarity Of Presentation:** 3
**Potential Impact:** 2
**Recommendation:** 3
**Confidence:** 5

**Review:**

This paper studies an important problem: how to connect different domains and embodiments to have an interface for general-purpose robot imitation learning, and proposes a partial image to flow to action framework to realize it. The structure of this paper is generally clear.

However, the originality of this idea is limited, since using flow as the interface has already been proposed by several recent papers. The main contribution of this paper is a new flow-generation network and flow-conditioned policy network, but these two networks lack enough baselines and comprehensive experimental validation to convince the readers the superiority of them compared to previous ones. Besides, the technical details of this paper are confusing.

Strength:
1. an important problem for general-purpose robot imitation learning
2. a new framework for generating flows and training a flow-conditioned policy

Weaknesses:
1. The motivation of some key network designs is missing
2. The originality of using flow as the interface is limited
3. The experimental comparison to previous flow-based methods is missing

**Quality Of The Limitations Section:**

1

**Questions For Rebuttal:**

1. Why should we use Stable Diffusion for flow generation? To my knowledge, the latent space of SD is "image features" from internet-level pictures that mainly focus on the color, spatial structure, texture, etc., rather than "flow features" that consist of a matrix of coordinates. I think these two feature spaces are totally different. The author should give more explanation on this, and add an experiment to show the superiority of using SD (with Animatediff) models over using other generative models such as VAE, DiT, and so on.
2. Why did the author only fine-tune the decoder of the VQGAN?
3. The author only compares their model to some ablation versions of their own, ignoring comparisons to other baselines such as ATM, Track2Act, and GeneralFlow. The author should add these baselines in their experiments to show the superiority of their method.
4. Why did the author design a "full flow prediction network" and an "alignment network" instead of directly training a single network that predicts the future flow at a given current observation?
5. Is it necessary to incorporate the 3D point cloud of the object? This will bring burdens for the application since we need additional operations to get the 3D point cloud. The author should add experiments to show it is necessary to use the point cloud as part of the state.
6. This work only considers object flow. What about the tasks that require considering two objects together? I think restricting the flow on a single object rather than the full image space (as in ATM) is a more restricted idea.

**Robotics Focus:**

4

**Summary Of Paper:**

This paper presents Im2Flow2Act, a general framework for imitation learning on cross-domain and cross-embodiment demonstrations with image inputs. The authors design new flow-prediction and flow-conditioned policy modules and show the effectiveness of their methods on different real-world tasks.

**Summary Of Recommendation:**

I think the main contribution of this paper is a new flow-generation network and a new flow-conditioned policy network, but it lacks enough experiments to show the supeority beyond previous designs. If the author can add enough experiments to show the superiority of their method compared to previous works such as ATM, GeneralFlow, and Track2Act, I will imporve my score.

---

### Official Review · Reviewer_aaJx · 2024-07-05
**a decent idea but questions need to be addressed**

**Originality:** 2
**Technical Quality:** 3
**Clarity Of Presentation:** 4
**Potential Impact:** 3
**Recommendation:** 2
**Confidence:** 3

**Review:**

Strength:

1. Using flow as an action representation is quite interesting and can be a promising direction for leveraging human data.
2. The proposed temporal alignment module, and object-based flow, are reasonable technical contributions for the reported stronger performance.
3. The paper is well-written and easy to follow.

Weakness: Please see the questions section for more details.

1. The authors do not thoroughly discuss the potential limitations of using flows in policy learning.
2. The proposed method seems to have a very strong assumption: Task action can be represented by a single object, and that object can be correctly detected.
3. Certain recent work with a similar idea, i.e., any point trajectory modeling for policy learning, is mentioned but not discussed in necessary detail.
4. Certain design choices need better justifications, e.g., why generate the entire task flow and use a temporal alignment module to track the task progress? Why not just predict the flow for the next couple of timesteps?

**Quality Of The Limitations Section:**

1

**Questions For Rebuttal:**

1. Flows are inherently noisy, lack real groundtruth (e.g. Optical Flow), and may not contain full information of an intended action. This poses many concerns about this approach:
    - Does this method work on tasks that require high precision, such as insertion type of tasks?
    - What about tasks that require rotation actions, such as screwing? The flow will be difficult to imagine.
    - What about certain actions that require only the moving of robot arm without objects? For example, reaching a certain button.
    - How to deal with the inherent randomness of flows? There are at least two types of randomness: 1) object movements are non-determinant, which can results large changes in flow for the same robot actions. 2) The output of off-the-shelf flow models is also noisy.

2. The proposed method seems to have a strong assumption: Every task action can be represented by a single object, and that object can be correctly detected. This limits the types of tasks to which the proposed method can be applied. For example, tasks that require tool use usually involve the interactions of multiple objects. Locating the key object alone can be very difficult.

3. The authors mentioned the ATM paper (any point trajectory modeling for policy learning) but did not thoroughly address the difference between Im2Flow2Act and ATM. ATM also proposes a flow-like action representation but does not use object-based flow. Based on question 2, this could even be an advantage for ATM for being a more general framework.

4. The proposed method generates the entire task flow and proposes a temporal alignment module to determine the current task progress. Why not just predict the flow for the next few timesteps? Is there a particular reason for generating the complete flow? Moreover, generating a complete flow seems to be infeasible considering the complex interaction of different objects in a long horizon.

**Robotics Focus:**

4

**Summary Of Paper:**

The paper presents a control policy that uses flow as its action representation. This enables policy learning from both robot and human demonstrations. The authors conduct real-world experiments to verify the effectiveness of the method and compare against heuristics-based policies.

**Summary Of Recommendation:**

I find this idea of using flow as action representation quite interesting. But I have many concerns, I will increase my score if my concerns are addressed.

---

### Official Review · Reviewer_WWt7 · 2024-07-21
**Review for Subimission 238**

**Originality:** 3
**Technical Quality:** 3
**Clarity Of Presentation:** 3
**Potential Impact:** 3
**Recommendation:** 3
**Confidence:** 3

**Review:**

This paper uses flow/point tracking trajectories as the representation from demonstrations. It can bridge the visual gap between simulation and real-world data. The proposed method learns from both demonstrations with and without actions. Visual-only demonstrations are used to learn a flow prediction network that predicts the future object movement given task descriptions. Demonstrations with both observations and actions are also used to learn a policy network that predicts actions from the flow representation. Flow acts as a nice bridge for visual-only demonstrations and embodied demonstrations.

However, I’m interesting how this method would work on the following two cases:
Long-horizon tasks. Point tracking drifts for long-horizon videos and flow generation for low-horizon tasks are exponentially more difficult. How to handle tasks with longer horizon remains a question.
Contact-rich tasks. Flow/point trajectories only record the pose of the end effector but not the force applied. How to use this representation for contact rich tasks like wiping something is a interesting question.

**Quality Of The Limitations Section:**

2

**Questions For Rebuttal:**

I would like the authors to answer the two questions in the review part. And here are some more detailed questions:
L140: can grounding dino help locate one of two identical objects by description of location? how to get  object of interes? Is it specified?
L160: what does it mean by slightly finetune?
L185-187: how to select initial keypoints? Why do we need 3D keypoints for initial observation but only 2D keypoints for the future frames? Why not using 2D or 3D for all the frames?
L234-238: would like to see the performances with flow generation model trained only on simulation or real-world data separately. It would help validate the idea that connecting simulation and real-world data with flow representation helps policy learning.
Supp L31: “opening the drawer to different extents”. Are you using programmed actions for play data? Would like a clarification on how to get actions in play data for all the environments.

**Robotics Focus:**

4

**Summary Of Paper:**

This paper learns a text-conditional flow generation model to predict object-centric flow given task description and initial observation, and learns a policy network that utilizes the generated flow to predict actions that finish the task.

**Summary Of Recommendation:**

The flow representation is interesting but seems to have some inherent insufficiencies for more general policy learning. I would like to hear authors' insight on this.

---

### Author Rebuttal · Authors · 2024-08-09

Please find our rebuttal PDF in the zip file, which includes additional experiment and clarifications.

---

### Decision · Program_Chairs · 2024-09-04

**Decision:**

Accept

**Comment:**

The paper introduces a framework for imitation learning across different domains and embodiments using image inputs. Reviewers generally agree that the paper addresses a significant problem and presents an interesting framework. However, concerns are raised about the originality of using flow as an interface, as it has been explored in previous works. The reviewers also highlight the lack of comprehensive experimental comparisons and sufficient baselines to validate the effectiveness of the proposed networks. Technical details need further clarification, especially regarding the motivation behind certain design choices and the practicality of flow generation using Stable Diffusion. The paper is well-written, but there are strong assumptions and limitations, such as representing tasks with single object flows, that need to be addressed.
Following the rebuttal, two of the three reviewers engaged in discussions with the authors, and the majority of their concerns were addressed, with both leaning towards accepting the paper.